# Waste Clearance in the Brain and Neuroinflammation: A Novel Perspective on Biomarker and Drug Target Discovery in Alzheimer’s Disease

**DOI:** 10.3390/cells11050919

**Published:** 2022-03-07

**Authors:** Kazuhiko Uchida

**Affiliations:** 1Faculty of Medicine, University of Tsukuba, 1-1-1 Tennoudai, Tsukuba 305-8575, Ibaraki, Japan; kazuhiko.uchida@cbiri.org; Tel.: +81-29-853-3210; Fax: +81-50-3730-7456; 2Institute for Biomedical Research, MCBI, 4-9-29 Matsushiro, Tsukuba 305-0035, Ibaraki, Japan

**Keywords:** Aβ clearance, blood-brain barrier, glymphatic system, innate immunity, mild cognitive impairment

## Abstract

Alzheimer’s disease (AD) is a multifactorial disease with a heterogeneous etiology. The pathology of Alzheimer’s disease is characterized by amyloid-beta and hyperphosphorylated tau, which are necessary for disease progression. Many clinical trials on disease-modifying drugs for AD have failed to indicate their clinical benefits. Recent advances in fundamental research have indicated that neuroinflammation plays an important pathological role in AD. Damage- and pathogen-associated molecular patterns in the brain induce neuroinflammation and inflammasome activation, causing caspase-1-dependent glial and neuronal cell death. These waste products in the brain are eliminated by the glymphatic system via perivascular spaces, the blood-brain barrier, and the blood–cerebrospinal fluid barrier. Age-related vascular dysfunction is associated with an impairment of clearance and barrier functions, leading to neuroinflammation. The proteins involved in waste clearance in the brain and peripheral circulation may be potential biomarkers and drug targets in the early stages of cognitive impairment. This short review focuses on waste clearance dysfunction in AD pathobiology and discusses the improvement of waste clearance as an early intervention in prodromal AD and preclinical stages of dementia.

## 1. Introduction

Dementia is one of the most economically burdensome diseases in families as it is a long-term illness causing disability in daily life. The number of people with dementia worldwide doubled from 20.2 million in 1990 to 43.8 million in 2016 [1]. According to the World Alzheimer Report 2021, the number of people with dementia worldwide is 55 million and is estimated to reach 78 million by 2030 [2]. Alzheimer’s disease (AD) is the most common type of dementia, followed by vascular dementia, dementia with Lewy bodies, and frontotemporal lobe dementia.

Late-onset AD, which occurs in people aged 65 years and above, has continuous progression from the preclinical stage without clinical symptoms to the pre-dementia stage, mild cognitive impairment (MCI), and AD with disability in daily life. Amyloid beta (Aβ) plaques and tau phosphorylation in the intraneuronal neurofibrillary tangles (NFTs) are major and classic characteristics of AD pathology. Aβ aggregates are neurotoxic and induce neurodegeneration; however, a substantial body of evidence has recently indicated that AD has a heterogeneous etiology and multifactorial pathogenesis [3].

Numerous research and clinical trials targeting Aβ have been performed in recent decades to overcome this devastating disease. There are currently 126 agents in 152 ongoing clinical trials for AD therapeutics, and new types of disease-modifying drugs targeting the underlying pathobiology of AD are also being developed [4]. In 2021, aducanumab, an anti-Aβ human monoclonal antibody, was approved by the United States Food and Drug Administration under the accelerated approval pathway for early AD [5]. However, many clinical trials of immunotherapy-targeted Aβ have failed to develop effective and safe drugs, possibly due to the heterogeneity of etiology and pathobiology in AD.

Neurodegenerative diseases are characterized by the deposition of aggregated proteins such as Aβ, tau, α-synuclein, and the TAR DNA-binding protein of 43 kDa (TDP-43) in the brain [6,7]. In sporadic AD, the accumulation of Aβ starts more than 20 years before disease onset without any symptoms, owing to impaired waste clearance rather than an overproduction of the peptide [8,9,10]. Recent studies have shown that early intervention is required for providing effective therapy and preventing dementia [11]. Although interventions with disease-modifying therapeutics in the preclinical stages of AD might be most efficacious, there is an urgent need for an assessment tool to monitor disease progression prior to significant cognitive decline.

The glymphatic (glia-lymphatic) system was recently discovered to be a waste clearance system in the brain using perivascular drainage [12]. Brain wastes, including the aggregates described above, are thought to be eliminated from the parenchyma using both the glymphatic system [13] and blood–cerebrospinal fluid (CSF) barrier at the choroid plexus (CP). This mini-review describes the recent advances in AD pathobiology, focusing on waste clearance in the brain and possible strategies for the intervention and prevention of AD.

## 2. Multifactorial Pathobiology in AD

### 2.1. The Danger Signal Activates Neuroinflammation and Induces Pyroptosis

Late-onset AD is the most common type of dementia among elderly individuals. As observed in autopsied AD brains, Aβ, which is produced from the amyloid precursor protein (APP) by the sequential cleavage of β- and γ-secretases, forms senile plaques. It is a major pathogenic factor as its production increases in hereditary AD. An experimental mouse model overexpressing APP demonstrated the loss of synaptic function and AD-like pathophysiology. Aβ42 accumulates in the brain parenchyma, forming plaques, whereas Aβ40 primarily accumulates in the cerebral blood vessel walls, causing amyloid angiopathy. The balance between Aβ production and clearance determines the amount of Aβ accumulation in the brain. An excessive accumulation of Aβ in the brain is observed in AD, whilst Aβ in CSF and peripheral blood is decreased due to an impaired clearance from the brain. The clearance of both Aβ42 and Aβ40 was reduced to 30% in AD individuals compared to that in non-demented controls [10]. APP processing occurs in an age-dependent manner possibly due to the inflammatory response. The imbalance of Aβ clearance and production might contribute to Aβ accumulation in late-onset AD.

Recently, caspase-1 dependent programmed cell death associated with inflammation has been proposed as a new type of cell death, named pyroptosis, which is characterized by inflammasome-related cell death with membrane rupture and cytokine release [14]. Several lines of evidence indicated that Aβ aggregates activate the inflammasomes and induce pyroptosis, resulting in glial and neuronal cell death [15,16,17]. Aβ binds to and stimulates the pattern recognition receptors, including the nucleotide oligomerization domain-like receptor proteins, resulting in inflammasome formation by the recruitment of the apoptosis-associated speck-like protein which contains a caspase recruitment domain and procaspase-1. Activated caspase-1 cleaves pro-interleukin (IL)-1β and pro-IL-18 into their mature forms, and cleaves gasdermin D to generate a 31-kDa N-terminal fragment which forms plasma membrane pores.

Several factors other than misfolded protein aggregates such as Aβ and tau prime inflammatory responses and induce caspase-1 dependent pyroptosis in the brain. The priming signals include lipopolysaccharide, bacterial pathogens, toxins, and double-stranded DNA. These pathogen-and host-derived signals induce innate immunity and activate nuclear factor-kappa B (NF-κB) signaling, resulting in the production of Aβ and inflammatory cytokines [18,19]. Thus, the production of Aβ may be a reactive compensatory response to several “danger signals” for nerve cells, such as pathogen-associated danger patterns (PAMPs) and damage-associated danger patterns (DAMPs) in the brain due to aging [20].

The role of infection in AD has been debated for 30 years [21]. Recent advances in research have reignited interest and discussion regarding the role of infection in AD. Several types of virus and bacteria including herpesvirus and *Porphyromonas gingivalis* (*P. gingivalis*) have been detected in the brain and colocalized with Aβ plaque [22,23]. Herpes simplex virus 1 (HSV-1) is usually latent in many elderly brains but may be reactivated under certain conditions leading to neuroinflammation with an increased risk of AD [24]. Other herpes viruses, including HSV-2, human herpesvirus-3 (HHV-3), and HHV-6, have been investigated for their association with AD [25]. *P. gingivalis* is a key pathogen in periodontitis, and several reports suggest a role of *P. gingivalis* in AD pathogenesis [26,27,28]. Viral and bacterial pathogens can cross the blood-brain barrier (BBB), and microglia recognize these pathogens and become activated, thus releasing proinflammatory signals in the brain parenchyma. Several lines of evidence supporting crosstalk between the peripheral and central immune systems suggest a role of systemic inflammation in AD [29,30,31,32]. More studies are required to address the question of whether chronic infection has a role in the disease progression of AD and whether the modulation of the signaling pathway in inflammation improves cognitive function.

### 2.2. The Glymphatic System for Clearance of Brain Waste

Recent studies on brain waste clearance have revealed that the impaired glymphatic system and dysfunction of the blood–CSF barrier (BCSFB) and BBB resulted in the accumulation of waste following neuroinflammation in AD pathogenesis [13,33,34,35]. Nedergaard and colleagues demonstrated that CSF moves into the parenchyma through the glymphatic system [12,36]. They showed that CSF enters the parenchyma through the periarterial pathway and flows towards the veins. The movement of periarterial CSF into the parenchyma facilitates the clearance of waste, such as Aβ, in interstitial fluid (ISF) into the paravenous space. The interexchange of CSF and ISF is crucial for brain homeostasis and waste clearance in the parenchyma. CP secretes CSF into the cerebral ventricles. Located in the brain ventricles, CP forms a BCSFB consisting of tightly connected epithelial cells, pericytes, and stromal space surrounding the capillaries and connective tissue.

CSF fluxes by the glymphatic paravascular system are reduced by aging. In an analysis using radiolabeled Aβ, the influx and efflux of CSF decreased in aged mice compared to young mice [37]. Cerebral vasculature aging is associated with CSF–ISF flux in the glymphatic system [38]. Arterial stiffness with aging leads to a reduction in arterial pulsatility, which drives the periarterial CSF into the parenchyma. The genetic deletion of aquaporin-4 water channels, expressed at the vascular endfeet of astrocytes, reduces glymphatic function and Aβ clearance by approximately 50% [12]. Aging is partly associated with the loss of perivascular aquaporin-4 polarization, resulting in the impairment of CSF–ISF exchange [37]. Failure of the glymphatic system due to aging reduces waste clearance in the parenchyma and results in the accumulation of Aβ aggregates and phosphorylated tau (p-tau), which induce neuroinflammation [13,36,39,40].

### 2.3. BBB and Blood–CSF Barrier in AD Pathobiology

Healthy brain function is dependent on healthy cerebral blood vessels and blood flow which supply oxygen and glucose. Oxygen and glucose transport across the BBB are a neurovascular coupling mechanism. The BBB is formed by a layer of capillary endothelial cells coupled with tight junctions, pericytes, smooth muscle cells, and astrocyte endfeet. To maintain brain homeostasis, BBB separates the parenchyma from the blood and prevents the infiltration of pathogens, blood cells, and neurotoxic components into the brain.

The physiology of molecular transport across the BBB and its dysfunction in neurological disorders have been intensively studied. Recent neuroimaging studies using dynamic contrast-enhanced magnetic resonance imaging (DCE-MRI) to quantify the regional BBB permeability revealed BBB breakdown in the hippocampus of individuals with early cognitive dysfunction [41] and aging [42]. BBB breakdown contributes to cognitive decline in apolipoprotein E (*APOE*)-4 carriers by Aβ-independent pathology [43]. Increased BBB permeability allows the infiltration of blood-derived macromolecules, leukocytes, hemoglobin, and neurotoxic agents produced by the pathogens. Because of BBB breakdown, free radicals are produced and proteases are released from leukocytes into the parenchyma. These pathogen- and host-derived “danger” signals lead to neuroinflammation in AD pathobiology.

### 2.4. Microglial Activation for Waste Clearance in the Brain

Microglia are present in the brain, forming a major component of the innate immune system. Increasing evidence supports the role of peripheral and central innate immune responses in neurodegenerative diseases [44,45]. Microglia express pattern recognition receptors, which bind and respond to PAMPs and DAMPs, including Aβ species. Activated microglia reduce Aβ accumulation by phagocytosis in a complement protein-dependent manner. Depending on the circumstances, microglia activation demonstrates a divergent response. Beneficial outcomes by increased clearance of debris with no or limited damage to neurons and detrimental outcomes can be observed by producing pro-inflammatory cytokines, leading to chronic inflammation and neurodegeneration [46]. There is cumulative evidence on an innate immune response suggesting peripheral and central crosstalk in AD pathogenesis [47,48]. Systemic inflammatory events are associated with cognitive decline in several cohorts and brain-resident macrophages and microglia are activated with the release of cytokines by the peripheral administration of lipopolysaccharides in mice models [30,49]. Thus, there is crosstalk between the peripheral and central immune systems. The inflammatory signals communicate with the central nervous system across BBB and BCSFB. The breakdown of these barriers activates innate immunity and induces neuroinflammation as normal responses in the preclinical stages of AD.

Genome-wide association studies (GWAS) for late-onset AD have identified several variants associated with the immune system, lipid metabolism, and synaptic function. A series of GWAS studies identified susceptibility genes, and several of these genes are involved in the regulation of waste clearance systems (Figure 1). The glymphatic CSF–ISF flow is driven by CSF influx in the periarterial space of major cerebral arteries and efflux to the perivascular space of venules. CSF–ISF drains into the meningeal lymphatic vessels and cervical lymph nodes. Arterial pulsatility and CSF flow are believed to drive ISF into the perivenous space by the hydrostatic pressure and osmotic gradients. *APOE*4 is a major genetic risk factor for AD which leads to BBB dysfunction in mice models and humans. After identifying *APOE*4 as a genetic risk factor for AD, *CLU* (also known as *APOJ*), *CR1*, *PICAM*, *ABCA*7, and *CD*33 were identified [50,51,52]. Astrocytic water channel aquaporin-4 (AQP4) plays a pivotal role in the glymphatic system, and *AQP*4 variants were associated with Aβ burden and an increased risk of AD [53,54]. Several association studies on genetic variations of low-density lipoprotein receptor-related protein 1 (*LRP1*) suggested that *LRP1* variants may not influence AD risk [55] but were associated with the lipid levels [56]. Genetic meta-analysis revealed an association of genetic variations with the immunity and lipid metabolism, including *CD33* and sortilin-related receptor-1 (*SORL1*) (also known as the low-density lipoprotein receptor relative with 11 ligand-binding repeats, *LR11*) [57]. Triggering receptor expressed on myeloid cells 2 (*TREM2)* carrying a rare variant (p.Arg47His) which is associated with three- to four-fold increased risk of developing AD [58,59].

TREM2 is a membrane receptor on microglia which is important for phagocytosis that helps in clearing the pathogenic molecules. Alpha-2 macroglobulin (A2M), a protease inhibitor, is a major component of innate immunity and may function to bind protein aggregates in AD pathogenesis. There is not enough evidence showing an association of *A2M* polymorphism with AD risk; however, A2M is reported to be a marker for neuronal injury and tau pathology. Thus, GWAS studies support a close relationship of waste clearance systems with AD pathogenesis.

In the AD brain, Aβ deposits are observed in the parenchyma and cerebral vessels. Cerebral amyloid angiopathy (CAA) is characterized by the accumulation of Aβ along the vessel walls and is observed in 30–90% of individuals with AD [60]. In CAA vessels, Aβ accumulation is associated with smooth muscle cells and results in vessel wall thickening and rupture. This structural remodeling of vessels might accelerate BBB dysfunction and the impairment of the glymphatic pathway. Thus, vasculature modification with aging might be an initial sign of cognitive impairment, followed by the dysfunction of waste clearance and Aβ accumulation in the brain. These age-related changes are believed to promote AD pathology in the preclinical stages of and/or prodromal AD (Figure 1B).

The BCSFB is composed of tightly packed CP epithelial cells and capillaries that lack tight junctions and are permeable to proteins and other macromolecules. CP mediates the selective transport of molecules from the blood into the CSF and from CSF into the blood via the CP epithelial cells. BCSFB disruption is a brain pathology in aging and cognitive impairment [61,62,63]. Aβ is transported bound to LRP1 and the receptor for advanced glycation end-products in the BCSFB and BBB [63,64,65]. CP dysfunction leads to impaired Aβ clearance as observed in AD mouse models [66]. There is decreased CSF secretion and turnover in AD. Lower levels of Aβ and higher levels of total tau and p-tau in the CSF are associated with AD and MCI [67,68]. These data suggest that CP function and CSF flow are crucial for CSF–ISF exchange in waste removal from the brain.

The transcriptome analysis of the postmortem brains aged 50–80 years revealed that a set of genes associated with neuronal and axon development changed between the ages of 50 and 79 [69]. Significant differences in several gene and protein expressions were observed in the CP of AD [70]. Transthyretin (TTR) is a homotetrameric protein which is primarily produced in the liver and CP and is secreted in the periphery and CSF [71]. TTR is neuroprotective against Aβ neurotoxicity by binding to Aβ aggregation [72,73]. TTR is decreased in the CSF is of patients with AD, and its expression was observed in the cortex and hippocampus of AD mice models [71].

## 3. Biomarkers for the Waste Clearance Dysfunction in AD

AD is diagnosed by the evidence of significant cognitive decline from previous levels, and assessed using objective neuropsychological tests and observing patients requiring assistance with complex instrumental activities of daily life. Biomarkers are pivotal in the development of therapeutics for neurodegenerative disorders. Table 1 summarizes the potential biomarkers of AD and MCI. Neuropsychological cognitive tests, neuroimaging, and blood tests are routine clinical examinations for the diagnosis of dementia. Among them, blood tests are easy to perform and the most realistic for periodic examinations of health checks in terms of cost savings and convenience.

Blood-based biomarkers for cognitive impairment have been intensively developed in the last decade. The presence of classic AD biomarkers Aβ and tau in plasma have been reported in several studies [74,75,76,77,78,79,80,81,82,112]. The results of large cohort studies indicate that plasma Aβ is related to amyloid burden, assessed using amyloid positron emission tomography (PET), and plasma total tau and p-tau appear to be the best biomarkers in symptomatic AD. A combination of these blood-based biomarkers may predict the onset of AD; however, they are not surrogate endpoints for the treatment response because the reduction in PET amyloid deposition was not associated with an improvement of cognitive function in many clinical trials of Aβ immunotherapies.

In the early stages of cognitive impairment, lifestyle changes and interventions to reduce the risk factors are effective for the improvement or prevention of cognitive decline. The World Health Organization’s guidelines on the risk reduction of dementia provide evidence-based recommendations on lifestyle [113]. Physical activity, nutritional intervention, and the management of hypertension and diabetes mellitus are strongly recommended for adults with normal cognitive function to reduce the risk of dementia. To evaluate the effectiveness of these interventions, endpoints should be set; however, it is difficult to evaluate extremely mild changes in cognitive decline with no cognitive impairment in the preclinical stages. In the case of therapeutic intervention at an earlier stage of the disease, classic neuropsychological batteries are useless in detecting extremely slight changes in cognitive decline. Therefore, the development of surrogate biomarkers for the early detection of cognitive impairment is vital for reducing the risk of dementia.

The neuroimaging of waste clearance efficacy is a potential target for the development of surrogate biomarkers [114]. MRI revealed an impairment of the glymphatic pathway and putative meningeal lymphatic vessels in the aging human brain [115]. Arterial spin labeling perfusion MRI was used to determine the perfusion clearance rate in early AD compared with normal subjects for clinical application [110]. The visualization of waste clearance in the brain might be useful for evaluating disease progression in AD pathobiology.

The level of proteins involved in Aβ clearance, TTR, apolipoprotein A1 (ApoA1), and complement component 3 (C3) in the blood are decreased in MCI and AD, and composite markers consisting of these three proteins had a high clinical potential to discriminate cognitive impairment from non-demented individuals in a multicentered clinical study [93,94]. TTR (known as prealbumin) is a blood marker of nutritional status and plays a pivotal role in waste clearance and protecting the brain against neurotoxicity. TTR is a major Aβ-binding protein that inhibits aggregation and neurotoxicity [72,116,117], and can transport Aβ from the brain across BBB [118]. A lower expression of TTR in CP and the dysfunction of CP with impaired Aβ clearance were observed in the 3XTg AD mice model [66]. The levels of TTR in both CSF and plasma are decreased in AD [93,94,101,102,103,104].

Albumin, a long-term nutritional marker, interacts with Aβ [100] and inhibits fibril formation [119]. Circulating albumin binds to Aβ in plasma for Aβ clearance in the liver. To remove the peripheral albumin-bound Aβ, albumin can be replaced in plasma; this has been proposed for AD therapy. A multicenter, randomized, blinded, placebo-controlled clinical trial of plasma exchange for AD management by albumin replacement was performed [120].

Lipid metabolism is associated with the homeostasis and modulation of AD pathology. There is a genetic link between the risk factors for AD and variations of the genome involved in lipid metabolism, including *APOE*, *CLU/APOJ*, and *ABCA7*. Apolipoproteins, such as ApoA1, ApoJ, and ApoE bind to Aβ and remove it via the BCSFB transporter LRP1. ApoA1 prevents Aβ aggregation and reduces its toxicity in primary brain cells as observed in AD mice models [121]. ApoE2 and E3 prevent Aβ aggregation, and apoE4 has a less effective inhibition of aggregation. ApoE transports Aβ across the BBB in an isoform-dependent manner (E4 > E3 > E2) [122]. ApoE and ApoJ are expressed in the brain parenchyma, and ApoA1 is transported from the periphery across the BBB and BCSFB. Peripheral lipid levels are associated with an incidence of AD. Higher LDL-C and TC concentrations were associated with an increased risk of AD [123]. A low plasma ApoE level is associated with a high risk of AD, as observed in 106,562 and 75,260 individuals in the general population [124]. A systematic review and meta-analysis revealed lower serum levels of ApoA1 in AD than in cognitively normal individuals [95]. Lower plasma ApoA1 levels are associated with an increased risk of clinical progression in *APOE*4 carriers with subjective cognitive decline [125].

In the healthy brain, complement activation is a prominent event in waste clearance by microglia [97]. Aβ aggregates are removed by microglia via an innate immune response in the brain. Both classic and alternative pathways of the complement system are associated with the pathogenesis of AD. C1q and C3 play a substantial role in the removal of Aβ deposits in AD pathology. Microglial cells are activated by the binding of complement proteins to their surface receptors. Microglial complement receptor 3 (CR3) regulates the brain’s Aβ levels. In the peripheral circulation, Aβ is captured by erythrocytes and macrophages in a complement-dependent manner, and CR1 on erythrocytes plays a role in Aβ clearance [126,127]. In individuals with MCI, peripheral levels of the native (inactivated) form of C3 and activated C3 fragments were found to be lower and higher, respectively, than in non-demented individuals [93,94]. A large cohort study revealed that low plasma C3 is a risk factor for AD in *APOE*4 carriers [128].

The blood-based biomarkers for AD and MCI can be categorized into three groups: protein nutrition, lipid metabolism, and innate immunity (Figure 2). These biomarker proteins function to eliminate Aβ from the brain and reduce its toxicity and damage to the neuronal cells. ApoA1 maintains the integrity of blood vessels, suggesting a link to vascular diseases. An improved management of cardiovascular risk factors such as hypertension, high cholesterol, and a reduced risk of stroke and heart disease have decreased the prevalence of dementia [129]. The brain vasculature drives glymphatic clearance in the parenchyma; therefore, aging-related vascular pathology is closely connected to AD. Low levels of ApoA1 are associated with type-2 diabetes mellitus [130,131], and the prospective population-based Rotterdam Study indicated that ApoA1 is a risk factor for type-2 diabetes mellitus [132]. Thus, a link exists between the risk factors for dementia and blood-based biomarkers, which contribute to the progression of vascular disease.

Biomarkers involved in waste clearance and waste neurotoxicity suppression are categorized into protein nutrition, lipid metabolism, and innate immunity biomarkers. These biomarker proteins are closely related to lifestyle-related diseases such as diabetes and hypertension, which are major risk factors for dementia in middle-aged and elderly individuals. The loss of function of these proteins affects the vasculature in the brain and reduces waste clearance. An increase in these biomarker proteins induced by one’s changing lifestyle as an intervention may improve brain homeostasis and inhibit neuroinflammation. The utilization of the blood biomarkers involved in waste clearance may play a critical role in dementia prevention.

## 4. Future Perspectives

How can we improve waste clearance in the brain? Figure 3 illustrates images of disease progression in AD with the involvement of “danger signals”, neuroinflammation, pyroptosis, and neuronal cell death. Regarding the prevention of dementia, waste clearance is crucial to maintain brain homeostasis and suppress neuroinflammation. It is still unclear whether monoclonal Aβ antibodies are effective in the treatment of preclinical AD. However, there are various approaches other than the removal of Aβ by passive immunotherapy. There are 126 agents in 152 trials assessing new therapies for AD: 28 treatments in phase 3 trials; 74 in phase 2; and 24 in phase 1 [4]. It should be noted that 61% of these treatments are disease-modifying therapies in Phase 3 trials, and among 17 trials, 11 are approaches other than Aβ and tau. They are small molecules, including drugs that block *P. gingivalis* toxicity and reduce the bacterial load, and inhibit ERK and NF-κB in inflammatory signaling pathways. In the preclinical stages of AD, neuroinflammation-target approaches, the protection of BBB integrity, and promotion of waste clearance have the potential to open a window of therapeutic opportunity in the near future. If the efficacy of such therapies could be monitored using biomarkers, we could improve cognitive decline and prevent the onset of AD.

## Figures and Tables

**Figure 1 cells-11-00919-f001:**
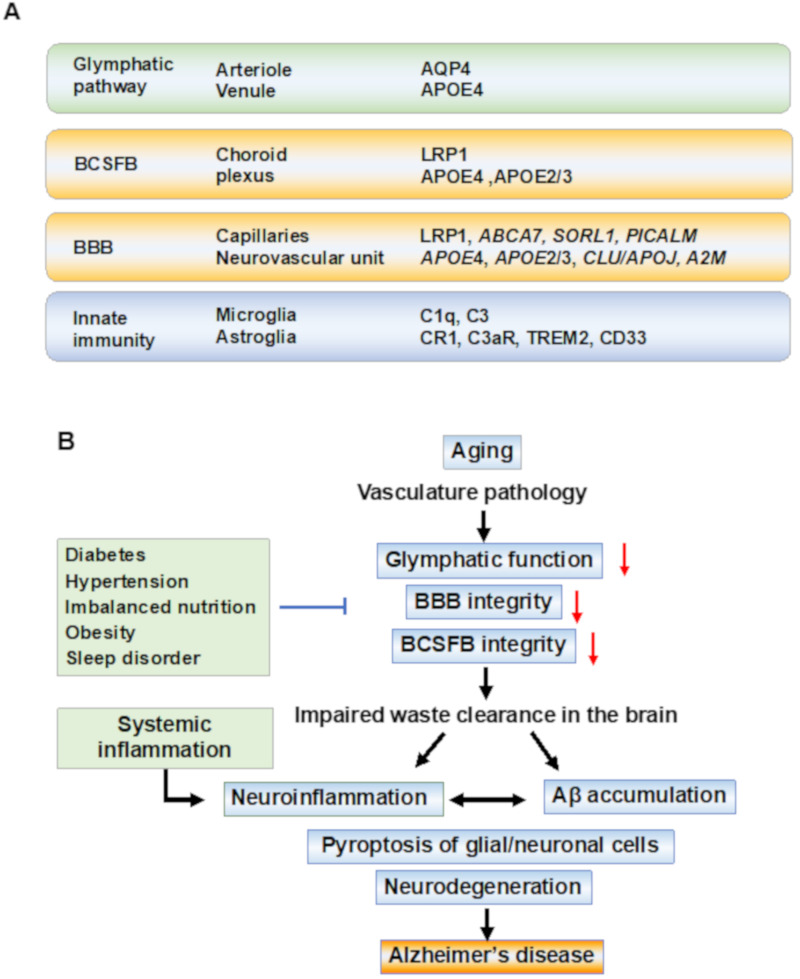
Waste clearance systems and pathobiological events in Alzheimer’s disease. (**A**) Anatomical regions and key molecules involved in waste clearance systems. Anatomical regions and key molecules identified in the genome-wide association study (GWAS) associated with Alzheimer’s disease (AD) are shown. The molecules which have genetic variations associated with an increased risk for AD are shown as identified by GWAS. The paravascular glymphatic flow is dependent on the water channel aquaporin-4 (AQP4) at the astrocytic endfeet. The blood–cerebrospinal fluid barrier (BCSFB) at the choroid plexus and the blood-brain barrier (BBB) at the capillaries regulate the transport of molecules into and out of the brain. Microglia survey the brain to support homeostasis, and the pathogens and amyloid-beta (Aβ) aggregates are cleared by microglia through receptor-mediated phagocytosis. Astrocytes (astroglia) promote neuroinflammation. After exposure to Aβ, astroglia release NF-kB and complement components, which act through the complement receptors such as complement receptor 1 (CR1) and complement component 3a receptor (C3aR). Aβ is removed by the low-density lipoprotein (LDL) receptor-related protein-1 (LRP-1)-mediated transport. The variants of adenosine triphosphate-binding cassette transporter A7 (ABCA7), sortilin-related receptor-1 (SORL1), phosphatidylinositol-binding clathrin assembly (PICALM), clusterin (CLU/APOJ), and alpha-2 macroglobulin (A2M) affect BBB and clearance functions. Among the rare variants of genes associated with an increased risk of AD, triggering receptors expressed on myeloid cells-2 (TREM2), CD33, and CR1 are expressed on microglia and ascribed to innate immune pathways. (**B**) Underlined mechanisms and pathobiological events in the disease progression of AD. In sporadic AD, the aging of the vasculature may be an initial event in AD pathobiology. The waste clearance pathways contribute to brain homeostasis and its dysfunction is associated with vasculature disease. In preclinical stages, age-related vascular changes including vasomotor dysfunction, structural remodeling, and chronic inflammation, cause the breakage of tight junctions between epithelial cells in the BBB, and barrier dysfunction in the brain leads to the neuroinflammation of the parenchyma. Inflammation-related signal transduction induces beta-site amyloid precursor protein (APP) cleaving enzyme-1 (BACE1) expression, leading to Aβ production following tau pathology, resulting in amyloid plaques and neurofibrillary tangles. The accumulation of Aβ reduces capillary blood flow and impairs waste clearance. Systemic inflammation is a risk factor for AD, and recent studies have suggested that systemic inflammation can drive neuroinflammation. In these aging processes, an increase in damage-associated danger patterns, including Aβ, which has synaptotoxicity in the brain, promotes neuroinflammation in the parenchyma, and finally, caspase-1 dependent programmed cell death (pyroptosis) in the glial and neuronal cells occurs, resulting in cognitive impairment. Abbreviations: AD—Alzheimer’s disease; BBB—blood-brain barrier; BCSFB—blood–CSF barrier.

**Figure 2 cells-11-00919-f002:**
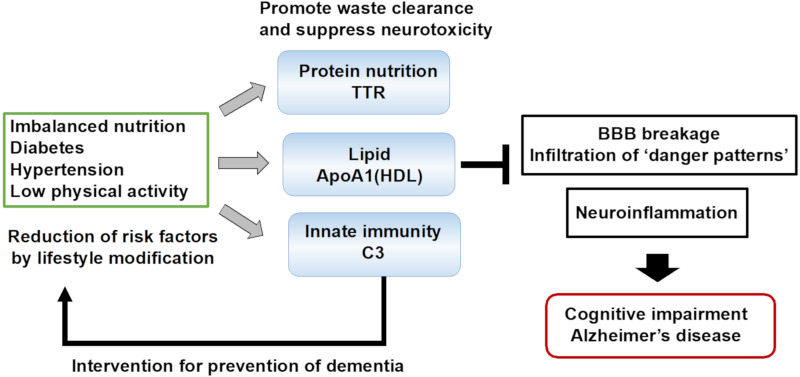
Possible roles of blood biomarker proteins in dementia prevention.

**Figure 3 cells-11-00919-f003:**
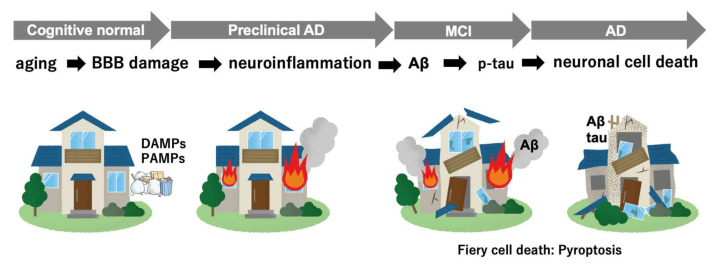
Illustration of disease progression from preclinical stages to onset of Alzheimer’s disease. DAMPs—damage-associated molecular patterns; PAMPs—pathogen-associated molecular patterns.

**Table 1 cells-11-00919-t001:** Potential liquid-based and neuroimaging biomarkers for Alzheimer’s disease.

Biomarker	Mechanism	Method/Target	References
Aβ	aggregation neurotoxic	PET CSF plasma EVs	[74,75] [76,77] [78,79]
Total tau P-tau	aggregation neurotoxic	PET plasma CSF EVs	[80,81,82]
NFL GFAP	neurodegeneration	serum/plasma CSF	[83,84,85,86,87]
miR-132, miR-124 miR-132, miR-146 miR-1908, miR-205	Aβ and tau pathology neuroinflammation cholesterol metabolism	serum EVs	[88,89,90,91,92]
ApoE ApoA1 ApoJ	lipid metabolism blood–CSF barrier integrity waste clearance	serum/plasma CSF	[65,93,94,95,96]
C3 C5 Clusterin	innate immunity waste clearance	serum/plasma CSF	[93,94,97,98,99]
TTR Albumin	blood–CSF barrier waste clearance	serum/plasma CSF	[93,94,100,101,102,103,104]
BBB breakage	BBB waste clearance	DCE-MRI	[41,42,43,105,106]
CSF–ISF exchange	glymphatic system waste clearance	gadolinium-enhanced glymphatic MRI DTI MRI DTI-ALPS ASL-MRI	[107,108,109,110,111]

Abbreviations: Aβ—amyloid-beta; ApoE—apolipoprotein E; ASL-MRI—arterial spin-labeled perfusion magnetic resonance imaging; BBB—blood-brain barrier; C3—complement component 3; CSF—cerebrospinal fluid; DCE-MRI—dynamic contract-enhanced magnetic resonance imaging; DTI-ALPS—diffusion tensor image analysis along the perivascular space; EVs—extracellular vesicles; GFAP—glial fibrillary acidic protein; ISF—interstitial fluid; NFL—neurofilament light chain; TTR—transthyretin. Representative mi-RNAs are shown.

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
