# Peer review of "Waste Clearance in the Brain and Neuroinflammation: A Novel Perspective on Biomarker and Drug Target Discovery in Alzheimer’s Disease"

_cells, 2022, doi:10.3390/cells11050919_

Round 1

Reviewer 1 Report

The author focused on the review of Waste Clearance in the Brain and Neuroinflammation as a Novel Perspective on Biomarker and Drug Target Discovery in Alzheimer's disease.

I have carefully read the manuscript that is comprehensive and well written. I would recommend this manuscript publish in Cells at this status.

Author Response

Thank you for your positive statement. I have revised the manuscript according to the Reviewers’ comments.

Reviewer 2 Report

The review of Uchida et al summarized recent findings on the “waste clearance in brain” in Alzheimer’s disease focusing in particular on biomarkers associated with defects in the clearance mechanisms. 
In summary the review is well structured, parts are divided by reasonable subheadings. Unfortunately, the depth of the review is not very balanced, making it suitable a readership that wants to have the essential aspects summarized but not for scholars working in this field who wants to have an update of current findings with detailed information.  
No doubt, the topic is in the clear focus of the journal and the special issue and facts and opinions are sufficiently separated and distinguishable from each other. The conclusion and summary is balanced and supported by the reviewed literature. 
Regarding the comprehensibility and the language, I have only minor criticism, however the manuscript could benefit from the proofreading of a native speaker.

I have some issues, which should be addressed

1) Figure 1 simplifies the underlying molecular mechanism in a way that it could be discussed very controversially. In particular it shows a feed forward / futile cycle in which Amyloid burden affects aging which e.g. affects vascular inflammation. Per definition aging is a physiological process and aging per se does not include pathological processes but imcrease the vulnerability of some age-related processes. Same for figure 3 which is too descriptive and simplistic.

2)    Instead of figure 1 I recommend a figure summarizing the molecular mechanisms associated with clearance and Alzheimer´s disease

3)    Please be more precise and detailed. The author refers to several aspects and papers without explaining the mechanistical relevance. Table 1 is extremely crude. E.g. which miRNAs are involved etc.

4)    The author completely neglects lipids as biomarkers: In particular in the last decade lipids have been shown to play a crucial role in the progression of Alzheimer´s disease. They are known to affect APP-processing  and Abeta aggregation themselves but are also crucial breakdown products of neurons accumulating during disease progression. This important part should be included in the review, which is, together with point 3 my major criticism. 

In my opinion this review is more a short review or an overview of an important topic. It lacks details and a comprehensive literature search. In my opinion it is fine, after addressing the above mentioned points, to publish it as a mini review. Otherwise for a normal review more efforts must be made to achieve sufficient depth and completeness.

Author Response

I appreciate your suggestions, which have been very helpful in improving the manuscript. All the comments I received have been taken into account for improving the quality of the manuscript.

1) Figure 1 simplifies the underlying molecular mechanism in a way that it could be discussed very controversially. In particular it shows a feed forward / futile cycle in which Amyloid burden affects aging which e.g. affects vascular inflammation. Per definition aging is a physiological process and aging per se does not include pathological processes but increase the vulnerability of some age-related processes. Same for figure 3 which is too descriptive and simplistic.  2) Instead of figure 1, I recommend a figure summarizing the molecular mechanisms associated with clearance and Alzheimer´s disease

Ans: Thank you for your suggestion. I have revised Figure 1 in the revised manuscript. Original Figure 1 has been deleted. A new figure that summarizes the waste clearance mechanisms associated with AD is designated as Fig 1A, and the corrected version of the original Figure 1 is designated as Fig 1B. (page 5). The revised manuscript includes the microglial role in the clearance of Aβ aggregates in the brain parenchyma and GWAS studies related to the waste clearance to address the molecular mechanisms (page 4, lines 162 - page 6, 245).

3) Please be more precise and detailed. The author refers to several aspects and papers without explaining the mechanistical relevance. Table 1 is extremely crude. E.g. which miRNAs are involved etc.

Ans: Thank you for your suggestion. In the revised manuscript, waste clearance mechanisms by the innate immunity, and information regarding the evidence from GWAS studies have been added (page 4, lines 180 - page 6, 245). I apologize for the incomplete description in Table 1. I have surveyed the literature for micro-RNA and representative micro-RNA, which are associated with cellular function, and a few of them were selected and indicated in Table 1 (page 7).

4) The author completely neglects lipids as biomarkers: In particular in the last decade lipids have been shown to play a crucial role in the progression of Alzheimer´s disease. They are known to affect APP-processing and Abeta aggregation themselves but are also crucial breakdown products of neurons accumulating during disease progression. This important part should be included in the review, which is, together with point 3 my major criticism.

Ans: Thank you for your constructive criticism. As the Reviewer indicated, lipids are crucial in the pathobiology of disease progression in Alzheimer’s disease (AD). Apolipoproteins are a component of lipids; for example, apoA1 is a major component of HDL. We reported that serum apoA1 was decreased in MCI and AD compared with non-demented controls in a previous study (References 94 and 95). In the revised manuscript, a paragraph was added to describe the importance of lipids in AD pathology. (page 8, lines 333 – page 9, 345)

In my opinion this review is more a short review or an overview of an important topic. It lacks details and a comprehensive literature search. In my opinion it is fine, after addressing the above mentioned points, to publish it as a mini review. Otherwise for a normal review more efforts must be made to achieve sufficient depth and completeness.

Ans: Thank you for your suggestion. As the Reviewer indicated, the present paper describes a holistic view on the role of the brain waste clearance system in the AD pathogenesis as a short review. I have revised the sentence in Abstract (page 1, line 23) and Introduction (page 2, line 65)

Reviewer 3 Report

This is an interesting paper with modern view on AD pathology. However, I have several comments:

  1. Authors claim that: Decreased clearance, rather than overproduction might contribute to Aβ accumulation in late-onset AD. In my opinion this statement is not completly true and this part should be rephrased.
    Indeed, in aging body decreased of microglia ability to correct Aβ clearance or pathogens phagocytosis lead to inflammatory response that switch from beneficial to detrimental (chronic inflammation). However, still present acute/chronic infections lead to Aβ overproduction, what causes over-deposition as senile plaques with lose/limited antimicrobial capacity of Aβ. Aβ deposits as well as different microbes, and their products infiltrating into the brain, might be an initiating factor of neuroinflammation and neurodegenerative changes observed in AD. Consequently, clinical interventions (clinical trials against Aβ) with Aβ reduction might have a catastrophic repercussions, such as encephalitis. 
  2. What about viral danger signals that activate neuroinflammation? It wasn't mentioned.
  3. While there is still no consensus regarding the role of infection in AD, studies and leading theories have identified at least three distinct frameworks that may ultimately characterize the role of chronic infection in AD. Maybe this paper would help: Butler L, Walker AK. The Role of Chronic Infection in Alzheimer’s Disease: Instigators, Co-conspirators, or Bystanders? Curr Clin Microbiol Rep. 2021.

Well done figures. Good luck with the review!

Author Response

1. Authors claim that: Decreased clearance, rather than overproduction might contribute to Aβ accumulation in late-onset AD. In my opinion this statement is not completely true and this part should be rephrased.
Indeed, in aging body decreased of microglia ability to correct Aβ clearance or pathogens phagocytosis lead to inflammatory response that switch from beneficial to detrimental (chronic inflammation). However, still present acute/chronic infections lead to Aβ overproduction, what causes over-deposition as senile plaques with lose/limited antimicrobial capacity of Aβ. Aβ deposits as well as different microbes, and their products infiltrating into the brain, might be an initiating factor of neuroinflammation and neurodegenerative changes observed in AD. Consequently, clinical interventions (clinical trials against Aβ) with Aβ reduction might have a catastrophic repercussions, such as encephalitis.

Ans: Thank you for these comments. It is difficult to address the question of whether an increase in production or a decrease in waste clearance determines Aβ levels. The balance between Aβ production and clearance determines the amount of Aβ accumulation in the brain. Therefore, I rephrased the sentences as per your suggestions (page 2, lines 79-83). Regarding the causative events, I believe that promotion of clearance rather than inhibition of production might be beneficial for reducing the AD risk because clinical trials on BACE inhibitors have failed due to efficacy and safety issues. Further studies are required to clarify these issues.

2. What about viral danger signals that activate neuroinflammation? It wasn't mentioned.

Ans: Herpes simplex virus (HSV) is known to be present in the brain, and several reports suggested a relationship between HSV and AD pathogenesis. This virus is usually latent in most elderly brains but may be reactivated under certain conditions, leading to neuroinflammation. The revised manuscript describes the possible roles of viral infection in neuroinflammation (page 3, lines 104-119).

3. While there is still no consensus regarding the role of infection in AD, studies and leading theories have identified at least three distinct frameworks that may ultimately characterize the role of chronic infection in AD. Maybe this paper would help: Butler L, Walker AK. The Role of Chronic Infection in Alzheimer’s Disease: Instigators, Co-conspirators, or Bystanders? Curr Clin Microbiol Rep. 2021.

Ans: Thank you for your suggestion and for providing essential literature to improve my paper. As the Reviewer indicated, the ‘infection theory’ on AD is still controversial. Recent advances in research reignited interest in the role of infection in AD. The suggested reference and paragraph regarding the role of infection in AD pathogenesis are added in the revised manuscript (page 3, lines 104-119, Reference 25).

Round 2

Reviewer 2 Report

The authors have adequately addressed my concerns. In my opinion, the manuscript is suitable for publication in its present form now